# Reinforcement Learning for Control of Non-Markovian Cellular Population Dynamics

**Josiah C. Kratz**[*]
Computational Biology Department
Carnegie Mellon University
Pittsburgh, PA 15213, USA
jkratz@andrew.cmu.edu

**Jacob Adamczyk**[*]
Department of Physics
University of Massachusetts Boston
IAIFI
Boston, MA 02125
jacob.adamczyk001@umb.edu

## Abstract

Many organisms and cell types, from bacteria to cancer cells, exhibit a remarkable ability to adapt to fluctuating environments. Additionally, cells can leverage a memory of past environments to better survive previously-encountered stressors. From a control perspective, this adaptability poses significant challenges in driving cell populations toward extinction, and thus poses an open question with great clinical significance. In this work, we focus on drug dosing in cell populations exhibiting phenotypic plasticity. For specific dynamical models switching between resistant and susceptible states, exact solutions are known. However, when the underlying system parameters are unknown, and for complex memory-based systems, obtaining the optimal solution is currently intractable. To address this challenge, we apply reinforcement learning (RL) to identify informed dosing strategies to control cell populations evolving under novel non-Markovian dynamics. We find that model-free deep RL is able to recover exact solutions and control cell populations even in the presence of long-range temporal dynamics. To further test our approach in more realistic settings, we demonstrate robust RL-based control strategies in environments with measurement noise and dynamic memory strength.

## 1 Introduction

In order to survive, organisms must adapt to unpredictable environmental stressors occurring over diverse timescales. As a result, biological systems display remarkable adaptive capabilities, making it exceptionally challenging to control them through environmental modulation, marking an open and significant question in the physics of living systems. Two examples of adaptive systems with great clinical importance are cancer cell resistance to chemotherapy (Vasan et al., 2019), and bacterial resistance to antibiotics (Blair et al., 2015). In both settings, the control problem of drug application is complicated by unknown model parameters and non-Markovian dynamics. Notably, the constant application of a drug does not typically result in population extinction, as drug application also drives a certain fraction of the population to alter their phenotypic state to become drug-resistant. This new resistant state can then persist even after drug removal and across cell lineages, thus encoding a memory of the stressful environment which allows populations to more quickly adapt if the drug is reapplied Harmange et al. (2023); Mathis & Ackermann (2017); Banerjee et al. (2021). As a result, it is unclear how to devise a control strategy that can effectively mitigate long-term population growth in these scenarios. We address this open problem in the present work.

---

[*]Equal contribution. Author ordering determined by coin flip at Primanti Bros.

Previous literature (Padmanabhan et al., 2017; Engelhardt, 2020; Gallagher et al., 2024; Fischer & Bluethgen, 2024) has studied temporal drug dosing protocols to slow or prevent adaptation in various cancer or bacterial models using various methods. However, these models fail to account for the variety of adaptive timescales present in real biological systems. The presence of such memory effects greatly complicates the control problem, making the study of more realistic models imperative. Thus, to study the control of such memory-based adaptive systems, in this work we introduce a novel population model exhibiting phenotypic plasticity with non-Markovian dynamics. Prior work has shown that learning-based methods are able to form the basis of patient-specific treatment protocols (Miotto et al., 2016; Ahn et al., 2021). We show that insights from control theory can be coupled with deep reinforcement learning to discover treatment protocols which successfully prevent proliferation. We highlight our main contributions below:

MAIN CONTRIBUTIONS

1. We develop a novel memory-based model for phenotypic switching relevant for bacterial and cancerous population dynamics.
2. We pose the problem of optimal drug dosing in this model and provide insights on the solution from control theory.
3. We find a robust optimal policy with deep RL that is independent of memory strength, requiring only clinically-accessible observations.

In the following sections, we first introduce the new population model, derive bang-bang optimal control, and experimentally show that bang-bang control is required to find a high-performing policy. Because of the non-Markovian dynamics, we introduce a short history to the agent's state which we implement via framestacking. Using ideas from optimal control theory and deep reinforcement learning, we thus simplify the original control problem in a well-principled manner. We then apply standard and more advanced RL techniques to solve different versions of the posed problem.

## 2 PROPOSED MODEL AND APPROACH

In this section, we develop our novel non-Markovian model for cellular dynamics. These dynamics are based on a simple but general switching model (Figure 1) with switching rates that are dependent on drug concentration. We then introduce a physically motivated non-linear memory kernel, before discussing general approaches to the optimal control problem.

### 2.1 NON-MARKOVIAN PHENOTYPIC SWITCHING MODEL

To model the treatment response of an adaptive cell population, we use a general phenotypic switching model which captures the time evolution of a susceptible subpopulation, with size $S(t)$, and a resistant subpopulation, with size $R(t)$. (The notation for susceptible and resistant states should not be confused with state and reward in RL, for which we use lowercase letters.) Phenotypic switching models have been successful in describing a wide variety of biological scenarios, including development of persister cells and resistant cells in bacteria, and development of drug resistance in cancer cells (Balaban et al., 2004; Fischer & Bluethgen, 2024; Witzany et al., 2023; Kumar et al., 2019; Kratz & Banerjee, 2024). In our model, susceptible cells with a net growth rate $\kappa_S(u)$ ($\kappa_S(u) < 0$ corresponds to net cell

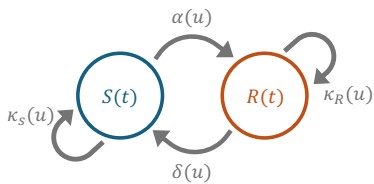

Figure 1: Depiction of the deterministic phenotypic switching model. The susceptible subpopulation $S$ transitions to the resistant state $R$ at a concentration-dependent rate, $\delta$. Similarly, the resistant state switches back to a susceptible state at a rate $\alpha$. The subpopulations each have a concentration-dependent growth or death rate, $\kappa$.

death) switch to a resistant state at a rate $\alpha(u)$, where $u$ is the drug concentration. Similarly, when the drug is removed, resistant cells with a net growth rate $\kappa_R(u)$ switch back to the susceptible state at a concentration-dependent rate $\delta(u)$ (Fig. 1). All growth and switching rates are a function of

drug concentration normalized by the maximum allowable dose, thus $u \in [0, 1]$. The time evolution of the size of each subpopulation, $\mathbf{x}(t) = [S(t), R(t)]^T$, is then given by the dynamical system:

$$\dot{\mathbf{x}}(t) = \mathbf{f}(\mathbf{x}(t), u(t)) = \mathbf{A}(u(t))\mathbf{x}(t) , \tag{1}$$

where the state-transition matrix is given by:

$$\mathbf{A}(u) = \begin{bmatrix} \kappa_S(u) - \alpha(u) & \delta(u) \\ \alpha(u) & \kappa_R(u) - \delta(u) \end{bmatrix} . \tag{2}$$

To relate the net growth rate to drug concentration for each subpopulation, we assume that both rates take the general form:

$$\kappa_S(u) = \kappa_S^{\max} - (\kappa_S^{\max} - \kappa_S^{\min})g(u) , \quad \kappa_R(u) = \kappa_R^{\min} + (\kappa_R^{\max} - \kappa_R^{\min})h(u) ,$$

where $g(u), h(u) \in [0, 1]$ is a monotonic dose-response function which relates drug dose to net growth rate, $\kappa_S^{\max} > 0$ denotes the maximum growth rate of the susceptible subpopulation in the absence of drug application, and where $\kappa_S^{\min} < 0$ is the maximum death rate of the susceptible subpopulation caused by application of the maximum drug dose ($u = 1$). Importantly, $\kappa_R^{\max} > 0$ denotes the maximum growth rate of the resistant subpopulation, which occurs in the *presence* of the maximum drug dose, and $\kappa_R^{\min} < 0$ corresponds to the maximum death rate of the resistant subpopulation, which occurs when the drug is removed. This parametrization corresponds to "drug addiction" behavior in the resistant subpopulation, a robust phenomenon which has been observed not only in cell culture (Suda et al., 2012; Sun et al., 2014; Moriceau et al., 2015), but in animal models (Das Thakur et al., 2013) and *in vivo* (Seifert et al., 2016; Dooley et al., 2016). Similarly, $\delta(u)$ decreases with $u$ while $\alpha(u)$ increases with $u$, as drug application drives the population to become more resistant, while reduction in drug concentration causes the system to recover susceptibility. These switching rates can then be defined as:

$$\alpha(u) = \alpha_{\max}j(u) \text{ and } \delta(u) = \delta_{\max}(1 - k(u)) ,$$

where $j(u), k(u) \in [0, 1]$ and where $\alpha_{\max}, \delta_{\max} > 0$ denote the phenotypic switching rates of cells switching from the susceptible state to the resistant state, and vice versa.

Recently, cell populations of many types, including human cancer cell lines, yeast, and bacteria, have been shown to maintain a memory of past environments which facilitates adaptation to previously-seen stressors over many timescales (Shaffer et al., 2020; Harmange et al., 2023; Larkin et al., 2024; Wolf et al., 2008; Mathis & Ackermann, 2017). To better capture this memory dependence on treatment response, we introduce a memory kernel into the previously-described dynamics (equation 1), making them non-local in time. Specifically, we choose a fractional differential equation (FDE) formulation as a phenomenological way to introduce multiple timescales of adaptation, one which has been used successfully to model memory effects in other biological (Lundstrom et al., 2008), ecological (Khalighi et al., 2022), and physical contexts (Bonfanti et al., 2020). With this addition the dynamics now become:

$$\dot{\mathbf{x}}(t) = \boldsymbol{F}(\mathbf{x}(t), u(t)) = \int_0^t \frac{(t - \tau)^{\mu-2}}{|\Gamma(\mu - 1)|}\mathbf{f}(\mathbf{x}(\tau), u(\tau))d\tau , \tag{3}$$

where $\Gamma(\cdot)$ denotes the Gamma function, $\mathbf{f}(\cdot)$ is given by equation 1, and here we introduce the parameter $\mu \in (0, 1]$ which controls memory strength. A value of $\mu = 1$ corresponds to the memoryless case (first order derivative), whereas smaller values of $\mu$ correspond to an increased influence of past states on the current dynamics (lower order fractional derivative).

We seek to obtain a temporal drug concentration protocol $u(t)$ which minimizes the growth of a population and thus choose the final cost as $C := \log N(T)/N(0)$ over the time interval $T$, where $N(t) = S(t) + R(t)$ represents the total population.

Thus, we aim to solve the following control problem:

$$\min_u C(\mathbf{x}(0), \mathbf{x}(T; u(\cdot))) \text{ subject to } \dot{\mathbf{x}}(t) = \boldsymbol{F}(\mathbf{x}(t), u(t)), \mathbf{x}(0) = \mathbf{x}_0, u(t) \in [0, 1] , \tag{4}$$

where $\mathbf{x}(T; u(\cdot))$ denotes the state of $\mathbf{x}$ at terminal time $T$ subject to control $u$ from $0 \leq t \leq T$. Interestingly, in our minimal model (equation 3), as long as $\kappa_S(u)$, $\kappa_R(u)$, $\alpha(u)$, and $\delta(u)$ are monotonic functions, we can show that the optimal control solution follows "bang-bang" control, regardless of the precise model parameters and memory strength.

## 2.2 DERIVATION OF BANG-BANG CONTROL

Using equation 1 and the fact that $N(t) = S(t) + R(t)$, the model dynamics can be rewritten as a single fractional differential equation in terms of the fraction of resistant cells, $\phi := R/N$, namely:

$$D_0^\mu \phi(t) = f(\phi, u) = (\kappa_S(u) - \kappa_R(u))\phi^2 + (\kappa_R(u) - \kappa_S(u) - \delta(u) - \alpha(u))\phi + \alpha(u) , \quad (5)$$

where $D_0^\mu$ denotes the Caputo fractional derivative of order $\mu$ starting at $t = 0$ (Garrappa, 2018), and here we drop the explicit time dependence for notational clarity. This can equivalently be written as the continuous delay differential equation (we use this form in equation 3):

$$\dot{\phi}(t) = \int_0^t \frac{(t-\tau)^{\mu-2}}{|\Gamma(\mu-1)|} f(\phi(\tau), u(\tau)) d\tau . \quad (6)$$

Given these definitions, the Hamiltonian associated with this control problem is then

$$H(\phi(t), u(t), \lambda(t)) = \lambda(t) f(\phi(t), u(t)) , \quad (7)$$

where the trajectory of the Lagrange multiplier $\lambda(t)$ is the solution to the costate equation (Gomoyunov, 2023):

$$\lambda(t) = -\frac{\partial_\phi C(\mathbf{x}(0), \mathbf{x}(T; u(\cdot)))}{\Gamma(\mu)(T-t)^{1-\mu}} + \frac{1}{\Gamma(\mu)} \int_t^T \frac{\partial_\phi \lambda(\tau) f(\phi(\tau), u(\tau))}{(\tau-t)^{1-\mu}} d\tau . \quad (8)$$

Applying Pontryagin's minimum principle, we obtain the following inequality

$$H(\phi^*(t), u^*(t), \lambda^*(t)) \leq H(\phi^*(t), u(t), \lambda^*(t)) , \quad (9)$$

which along with equation 5 and equation 7 can be used to obtain the optimal control policy:

$$u^* = \arg\min_u \lambda^*[\Delta_S(\phi^* - \phi^{*2})g(u) + \Delta_R(\phi^* - \phi^{*2})h(u) + \delta_{\max}\phi^* k(u) + \alpha_{\max}(1 - \phi^*)j(u)] ,$$

where the following shorthand notation is introduced: $\Delta_S = \kappa_S^{\max} - \kappa_S^{\min}$ and $\Delta_R = \kappa_R^{\max} - \kappa_R^{\min}$.

Critically, the fact that $\kappa_S^{\max}, \kappa_R^{\max}, \alpha_{\max}, \delta_{\max} > 0$, $\kappa_S^{\min}, \kappa_R^{\min} < 0$, and $g, h, j, k, \phi \in [0, 1]$ ensures that the bracketed sum which multiplies $\lambda^*$ is positive for $u > 0$. As a result, as long as $g(u), h(u), j(u),$ and $k(u)$ are monotonically increasing functions of $u$, then the resulting optimal control is said to be "bang-bang", where $u^*(t)$ only takes on its extreme values. Furthermore, the switching times between the maximum and minimum values of $u$ are determined by $\lambda^*(t)$, yielding the optimal control solution:

$$u^*(t) = 0 \text{ if } \lambda^*(t) > 0,$$
$$u^*(t) = 1 \text{ if } \lambda^*(t)) < 0, \text{ and}$$
$$u^*(t) \in [0, 1] \text{ otherwise.}$$

In principle, the optimal control trajectory can be obtained through numerical integration of the model dynamics (equation 5) along with the corresponding costate equation (equation 8). In the context of bang-bang control of non-Markovian systems however, using this approach can be difficult, as it requires careful choice of integration technique and update rule. Thus, we turn to reinforcement learning.

As shown above, the optimal control solution to Problem 4 follows bang-bang control, regardless of model parameters. This allows the continuous model of equation 1 to be simplified to a discrete model with binary controls without altering the optimal solution. This yields:

$$\dot{\mathbf{x}}(t) = \mathbf{f}(\mathbf{x}(t), u(t)) = \begin{cases} \mathbf{T}\mathbf{x}(t) & \text{for } u = 1 \text{ (Treatment Phase)} \\ \mathbf{P}\mathbf{x}(t) & \text{for } u = 0 \text{ (Pause Phase)} \end{cases} , \quad (10)$$

where now there are two state-transition matrices, given by:

$$\mathbf{T} = \begin{bmatrix} \kappa_S^{\min} - \alpha_{\max} & 0 \\ \alpha_{\max} & \kappa_R^{\min} \end{bmatrix}, \qquad \mathbf{P} = \begin{bmatrix} \kappa_S^{\max} & \delta_{\max} \\ 0 & \kappa_R^{\min} - \delta_{\max} \end{bmatrix} . \quad (11)$$

We use this formulation of the environment when training the reinforcement learning agent.

## 2.3 CONTROL STRATEGIES

Despite this simplification in action space (from continuous to binary controls), the optimal control problem remains difficult, as the number of times and duration of drug application must be optimized. Constant application of the drug at the maximum dose ($u = 1$) results in cell adaptation and proliferation (Fig. 3): a highly suboptimal solution to Problem 4 and a catastrophic result in the clinical context.

Recently,(Fischer & Bluethgen, 2024) has shown that in the memoryless case ($\mu = 1$), the optimal solution for this type of model requires an initial drug application phase, followed by pulsing between treatment and pause phases at a regular interval dependent on the model parameters. However, as seen in Fig. 3, we find that the addition of memory ($\mu < 1$) renders the control strategy of the memoryless case ineffective, as cells which have previously encountered treatment switch to the resistant state faster upon subsequent applications. Furthermore, in the clinical or experimental setting, obtaining the values which parameterize equation 3 is usually not feasible, and one can only rely on direct and more macroscopic measurements from the cell populations. Thus, obtaining the appropriate switching frequencies through direct computation via optimal control (OC) theory becomes impossible. One may propose to learn these values directly before feeding them into an OC solution, but this may result in a brittle pipeline, especially as the model discussed may not fully describe the underlying biological dynamics. As a result, we seek to learn the optimal policy end-to-end, directly through experience using deep reinforcement learning (Sec. 3).

## 2.4 OBTAINING THE OPTIMAL SOLUTION IN THE MEMORYLESS CASE

As mentioned above, previous work (Fischer & Bluethgen, 2024) showed that the optimal pulsing protocol for the memoryless case requires an initial drug application phase until the resistant fraction reaches some upper threshold $\phi_h$, followed by a pause phase ("drug holiday") in which the resistant fraction naturally relaxes back to some lower bound $\phi_l$. This is followed by similarly repeated cycles of treatment and pause phases where the switching time occurs when the resistant fraction reaches $\phi_h$ and $\phi_l$, respectively. To obtain the optimal values of $\phi_h$ and $\phi_l$ for our specific model parameterization, we swept over values of $\phi_h$ and $\phi_l$ between 0.1 and 0.9, with increments of 0.011, selecting the values which yielded the highest net death rate: $\phi_l = 0.494$ and $\phi_h = 0.505$.

To compare as baselines against our learned policy in non-Markovian environments, we repeated the same procedure for different values of $\mu$. The optimal switching fractions are given in Table 1.

## 3 REINFORCEMENT LEARNING

| $\mu$ | $\phi_l$ | $\phi_h$ |
|-------|----------|----------|
| 1.0   | 0.494    | 0.505    |
| 0.9   | 0.483    | 0.494    |
| 0.8   | 0.466    | 0.477    |
| 0.7   | 0.448    | 0.459    |
| 0.6   | 0.428    | 0.439    |

Table 1: Optimal switching fraction values for the baseline policy for each memory strength $\mu$.

Reinforcement learning (RL) allows an agent to learn an optimal control strategy (policy) directly from the experience it collects. This data may be from simulation (Akkaya et al., 2019), or from real-world data (Haarnoja et al., 2024), or even from fully offline datasets (Levine et al., 2020). In RL, the interactive agent receives observations from the environment. As a result of this input, the agent outputs a control or action (denoted $u$ throughout), yielding a transition to a new state and a scalar reward (negative cost). The goal of the agent, analogous to OC, is to act in a way that maximizes the expected long-term accumulated rewards. In our problem formulation, this will correspond to the use of drug to minimize the total population count.

We consider the online RL setting in simulation, where a drug protocol is learned through experience (Fig. 2, left) by continually improving a policy as it is learned. Despite the non-Markovian dynamics and even without access to the underlying environment-specific model parameters, we show that RL is capable of providing high-performing policies. To formulate the RL problem, we define the relevant characteristics as follows (in the following, let $\Delta$ denote the simulation time between actions):

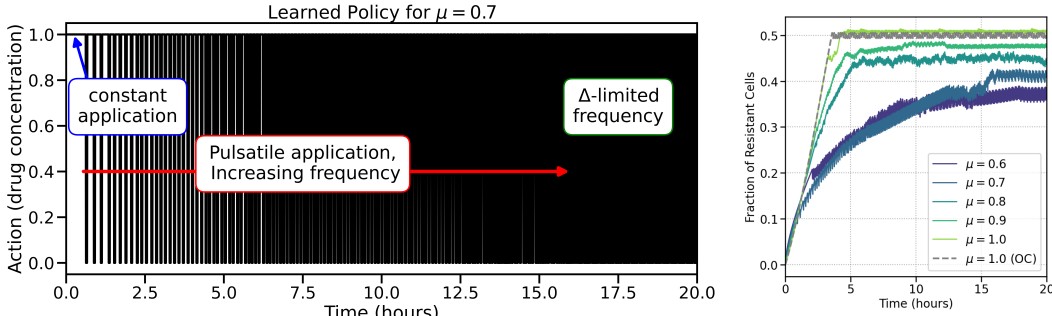

Figure 2: **Left**: The learned policy shows a resemblance to the optimal memoryless strategy, with an initial constant application phase followed by a pulsatile phase. However, in the case of memory-based dynamics, the frequency of pulsing must be increased over time as discussed in Sec. 4. Since the policy is eventually limited by the simulation time, the pulsing frequency becomes bottlenecked by our choice of time discretization $\Delta$ after $\approx 20$ hours. Despite this, the policy is still able to perform well with rapid pulsing. **Right**: Effect of learned policy on resistant fraction. for different memory strengths. The RL agent finds (for distinct $\mu$ values) appropriate lower and upper bounds for the fraction of resistant cells. Maintaining the subpopulation in this range ensures the population can be controlled.

**State**: The state $(s_t)$ of the agent's environment is a list of the last $K = 5$ estimates of the instantaneous growth rate and one-hot encoded actions, $s_t = [c_{t-K,..t}, u_{t-K,..t}]$ where $c_t = \Delta^{-1} \log N_t/N_{t-\Delta}$ is a pointwise version of the cost defined above. Thus, a history of past observations is encoded in the state vector, a crucial design choice if the agent is to learn control without a recurrent hidden state. **Action**: As motivated in Section 2.1, we choose a binary action space $u \in \{0, 1\}$ representing whether the drug is applied or not, as we have proven that this is sufficient to recover optimal control. **Reward**: As we seek to solve Problem 4, the reward is simply the negative growth rate, $r_t = -c_t$. Notice that with this choice of reward function, the sum of rewards across a trajectory simplifies as $R_{0:T} = \Delta^{-1} \log N_0/N_T$, ensuring the agent's objective is aligned with a reduction in total cell population. **Dynamics:** The dynamics of the total cell population $N_t$ are governed by the dynamical system described in Section 2.1, initialized to be fully susceptible $(\mathbf{x}_0 = [1000, 0])$. After an action is executed, the simulation (a numerical solution [1] of equation 3) is computed with the action (dose) fixed for $\Delta = 0.01$ hours to compute the next state $(s_{t+\Delta})$. To encourage the agent to reduce the cell population while decreasing computational runtime and allowing for exploration, we terminate the episode if the number of cells ever exceeds 150% of its initial amount.

The goal in RL is to learn the policy $\pi^*$ which maximizes the expected cumulative discounted sum of rewards across a trajectory:

$$\pi^*(u|s) = \arg\max_{\pi} \mathbb{E}_{\tau \sim p, \pi} \left[ \sum_{t=0}^{\infty} \gamma^t r(s_t, u_t) \right]. \tag{12}$$

To solve this optimization problem, we consider discrete-action value-based methods. Within this framework, the optimal action-value function $Q^*(s, u)$ is learned based on off-policy data observed during exploration. The value function satisfies the Bellman optimality equation (written to match the control notation introduced above):

$$Q^*(s_t, u_t; \theta) = r(s_t, u_t) + \gamma \max_{u'} Q^*(s_{t+\Delta}, u'; \theta). \tag{13}$$

 Within this class of methods, DQN (Mnih et al., 2015) learns $Q^*$ by exploring with an annealed greedy exploration strategy. After an action is taken, the experience tuple $(s, u, r, s')$ is stored in an experience replay buffer for later use. In deep RL, the action-value function is parameterized with a neural network (denoted by the trainable parameters $\theta$), and trained via mini-batch gradient descent,

---

[1]The numerical solution of fractional differential equations requires some care; cf. (Garrappa, 2018) for details.

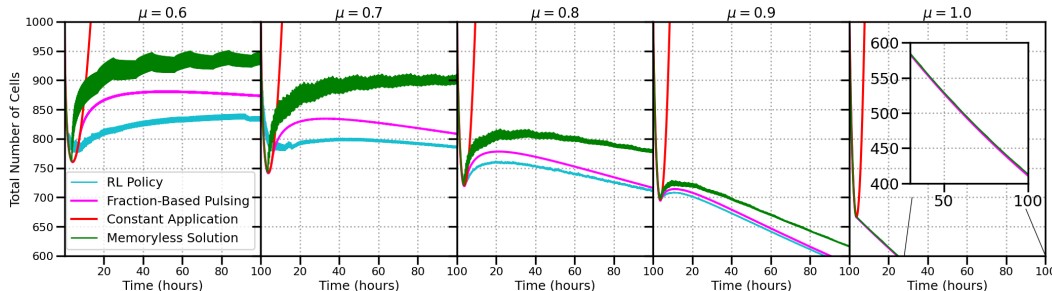

Figure 3: Performance comparison of constant drug application, solution for the memoryless case, resistant fraction-based pulsing technique, and policy learned by RL. For the fraction-based policy, an optimal lower and upper bound for resistant fractions are found through sweeping (Appendix 2.4). The RL policy is capable of controlling the cell population better than any other scheme.

denoted $\mathcal{B}$, uniformly at random from the buffer. The temporal-difference loss function is defined as the Bellman residual – the squared difference between left and right hand sides of equation 13:

$$\mathcal{L}_\theta = \sum_{\{s,u,r,s'\}\in\mathcal{B}} \left| Q^*(s_t, u_t; \theta) - \left( r + \gamma \max_{u'} Q^*(s_{t+\Delta}, u'; \bar{\theta}) \right) \right|^2 . \tag{14}$$

As common in value-based algorithms, we use an additional target network to reduce the effect of bootstrapping during training. The target network (denoted with parameters $\bar{\theta}$) has its parameters frozen during training, and updated periodically directly by copying the online network parameters $\theta \to \bar{\theta}$. We additionally use the double DQN approach to mitigate the over-estimation bias in standard DQN (Van Hasselt et al., 2016). Further details on reinforcement learning can be found e.g. in (Sutton & Barto, 2018; Hessel et al., 2018). We tune over several hyperparameters (whose values we list in the Appendix) to optimize performance. Our implementation of double DQN (Van Hasselt et al., 2016) (and later FQF) are based on open-source code from Stable-Baselines3 (Raffin et al., 2021). Each agent is trained for $3 \times 10^5$ environment steps. All code to reproduce our experimental results can be found at `https://github.com/JacobHA/RL4Dosing`.

## 4 CASE STUDY OF FIXED MEMORY ENVIRONMENTS

We first test DQN in the memoryless case ($\mu = 1$), for which an optimal controller is known (Fischer & Bluethgen, 2024). In this case, the optimal policy can be derived based only on two consecutive resistant fractions. However, recall that the RL agent is only given access to the instantaneous growth rates, not the resistant fractions. Nevertheless, the RL agent is surprisingly able to reliably recover the optimal policy with only two successive frames and without any access to underlying model parameters. The control strategy found by the agent involves first applying the drug and then pulsing regularly, in agreement with the OC solution, as seen in Fig. 3 (rightmost plot). Indeed, Fig. 2 (right panel) shows that the agent discovers an internal representation based on growth rates that is compatible with fraction-based switching.

With confirmation that RL can find the optimal dosing strategy in the memoryless case, we turn to the more difficult memory-based dynamics ($\mu < 1$). We do not have a solution to Problem 4 in this regime, so we compare against two baselines: The memoryless protocol described above and a modified version in which switching times occur when the resistant fraction reaches a threshold value. We note that these baselines can have arbitrarily small switching times, which is practically infeasible. Remarkably, we find that using a small but bounded $\Delta$, our learned policy outperforms both of these baselines (Fig. 3). In addition, our experiments show that decreasing $\Delta$ (increasing measurement frequency) beyond a certain threshold does not considerably increase performance (Fig. 6).

Interestingly, the learned policy reveals that the agent initially maintains constant drug application before transitioning to a memory-dependent pulsing protocol. The agent increases the frequency of pulsing throughout the trajectory until saturation at the maximum rate ($\Delta^{-1}$) (Fig. 2). For cases with memory, this increase in dosing frequency can be understood as the result of

faster cellular adaptation back to the resistant state after multiple drug encounters. Thus, to maintain a susceptible population, the policy's pulse frequency must continuously be increased to compensate for the memory-based adaptability. We also find that the agent maintains a lower average resistant fraction at higher memory strengths (Fig. 2, right). This qualitative trend is in agreement with the optimal switching fractions found by optimal control shown in Table 1.

To independently show the benefit of using discrete actions, the original problem can instead be approached with continuous action algorithms, such as PPO (Schulman et al., 2017) or SAC (Haarnoja et al., 2018). As shown in Fig. 4, these algorithms considerably underperform relative to DQN.

Qualitatively, we find that these continuous action algorithms successfully find the constant application period at the beginning of the episode. As the episode continues, though, both PPO and SAC attempt a short-lived bang-bang protocol before defaulting to the average action for the remainder of the episode. These results were obtained from an equal-compute hyperparameter sweep in a fixed $\mu = 0.9$ environment. We conjecture that continuous action algorithms struggle because of the saturating gradient effect at the limits of action space, which correspond to bang-bang control. Additionally, continuous action algorithms that give bonuses for policy entropy (e.g. SAC) make the task of learning these deterministic policies even more difficult.

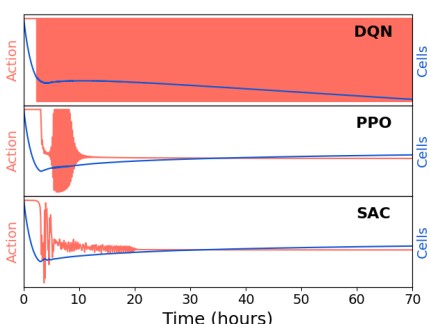

Figure 4: PPO and SAC fail to find a bang-bang control policy and have a lower performance than DQN; highlighting the need for discrete action algorithms, as informed by optimal control.

## 5 MODEL SUCCESSFULLY GENERALIZES TO MEMORY AND OBSERVATION NOISE PERTURBATIONS

In the previous sections, we construct RL agents that are each trained on specific dynamics determined by a fixed $\mu$ value. However, this can be problematic in practice, as the memory strength $\mu$ is difficult to measure (it would require fitting equation 5 to entire episodes of data), which could correspond to adverse costs in the clinical setting. Furthermore, adaptive cell populations such as cancer can dynamically change their memory capacity over time (Ringrose & Paro, 2004; Acar et al., 2005). Thus, a useful policy should be able to interact immediately without requiring additional samples, and with no access to the value of $\mu$. The unknown value of $\mu$ thus represents "side information" that is not given to the agent but provides a *context* for the agent to understand the current environment. This places the new problem setting in the framework of contextual MDPs (Hallak et al., 2015; Sodhani et al., 2022). To approach this problem while maintaining clinical applicability, we train a general agent over a range of memory strengths. This approach of "domain randomization" (DR) has been applied with great success in other RL problems (Tobin et al., 2017; Akkaya et al., 2019).

To tackle this more challenging environment, we use FQF (Yang et al., 2019), a recent algorithm for distributional RL, where the distribution (rather than only the mean) of future returns is learned. FQF and its variants have shown success in discrete control environments. In particular, we use a noisy variant that combines the approach of NoisyNets (Fortunato et al., 2018) for improved exploration. Further, we find it useful to supplement the RL state with the one-hot encoded actions used in the last $K$ steps. This allows the agent to observe the response of the environment to the presence of drug: At lower $\mu$, application of drug has a drastically different effect than at higher $\mu$ values at similar growth rates. With these changes, we sweep over hyperparameters (cf. Appendix) to find a model capable of reducing cell populations given any memory strength.

Although the $\mu$-specific models outperform the generalist agent in their own finetuned environments, the generalist agent can achieve good performance across a range of memory strengths (Fig. 5). It is worth noting that this model now enables use in settings where memory strength is entirely unknown

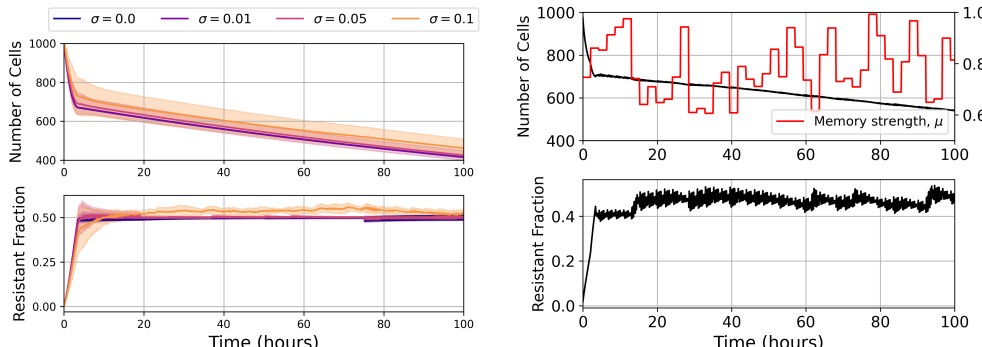

Figure 5: Left: Policy is robust to observation noise, as tested in a memoryless environment. To represent potential measurement errors in the clinical setting, noise is drawn from a normal distribution with standard deviation $\sigma$ and added to each state before given to the agent. Despite large amounts of observation noise, the agent was able to drive population reduction and maintain a similar resistant fraction. Each trace represents the mean over 10 trajectories, with the standard error represented by the shaded region. Right: Learned policy is general and robust to changes in memory strength. Every 20 decision steps the memory strength is reset to a new value, drawn from a uniform distribution over the interval $[0.6, 1]$. Agent is able to quickly adapt to mitigate population growth in the new environment.

and possibly dynamical or noisy in nature. Experimentally, we find this agent is very robust to perturbations of various types: namely large changes in $\mu$ and large amounts of observation noise. In both cases, the learned policy employed by the agent is able to successfully adapt and continue to minimize population growth (Fig. 5).

## 6   RELATED WORK

The use of reinforcement learning in control of biological systems, especially in dosage control, has been investigated by others. We will briefly highlight the most relevant prior work in this section. (Padmanabhan et al., 2017) studies a deterministic memory-less model of cancer dynamics with additional "immune" and "normal" cell types included. The number of tumor cells is used as a discrete RL state, and hence a tabular method is used. In (Engelhardt, 2020), a coupled SDE governs dynamics of $d$-many phenotypic subpopulations and the RL state is composed of measurements of individual phenotype populations. Although the specific dynamical model varies, prior work often requires unrealistic inputs to define the RL observation. An important contribution of the current work is in advancing the model class itself while maintaining clinical relevance by using easily measured quantities and providing a policy that generalizes across memory strengths.

## 7   FUTURE WORK

We utilize frame-stacking (Mnih et al., 2015) to encapsulate the history of the agent's trajectory, but in future work we will test the use of recurrent policies to capture more nuanced long-term effects which may further improve performance. We also plan to extend our deterministic framework to the stochastic setting, as population heterogeneity is known to further complicate the control process. This work demonstrates the benefits of combining OC and RL to simplify the control problem, showing how their frameworks can be effectively integrated. This connection can be further developed by using the memoryless OC solution to enhance RL training through reward shaping (Ng et al., 1999; Wiewiora, 2003; Adamczyk et al., 2023) or pre-training (Uchendu et al., 2023) techniques.

In this work we present a strong proof of concept for RL-guided drug dosing, which we hope will motivate further experimental work on adaptive therapies. Since preparing drug doses of arbitrary concentration can be challenging, the derived bang-bang control simplifies the practical implementation of our policies. While our results show that the optimality of such policies is robust against

memory strength, switching rates, and dose-response curves, their optimality in the clinical setting may be influenced by additional factors such as patient well-being, interactions with the host system, and penalties for drug exposure. Addressing these constraints highlights the need to bridge the "sim-to-real" gap through both novel computational approaches and evaluations on real-world data.

Inspired by its success in robotics applications (Akkaya et al., 2019), domain randomization may prove useful in addressing such computational challenges. By training agents in a diverse set of simulated environments with randomized parameters (e.g., switching rates, dose-response effects, noise levels), policies can become more robust for real-world application. With such general agents successfully trained, an offline RL problem (Levine et al., 2020) with warmstarting (Uchendu et al., 2023; Agarwal et al., 2022) could be formulated once real-world data is available. We hope that this approach can provide a path toward generalization, thus enabling more reliable application of RL policies in diverse real-world contexts.

## 8 DISCUSSION

In this work, we study the control of a highly non-Markovian model of adaptive cellular growth dynamics using deep reinforcement learning. Although we focus here on a specific parameterization most relevant to cancer, we expect this methodology to be applied successfully to other scenarios, including resistance development in bacteria. We find that deep RL is capable of recovering the known optimal policy for the memoryless case and can successfully find a policy for memory-based systems which prevents proliferation, all without access to the underlying model parameters. Furthermore, we obtain a policy that is robust to measurement noise and memory strength perturbations.

Exact methods from optimal control require direct access to the population's resistant fraction and underlying model parameters to determine switching rates. In practice, experimentally measuring the growth rate of a pathogenic population is simpler than determining its corresponding resistant fraction. Surprisingly, we find that model-free reinforcement learning can provide successful policies directly from the growth rate itself, making it a promising method for use in clinical settings.

## 9 ACKNOWLEDGEMENTS

JA acknowledges funding support the NSF through Award No. PHY-2425180; the use of the supercomputing facilities managed by the Research Computing Department at UMass Boston; and fruitful discussions with Rahul V. Kulkarni and Stas Tiomkin. JCK acknowledges support from the Computational Biology Department at CMU, and from Shiladitya Banerjee in the Department of Physics at CMU. This work is supported by the National Science Foundation under Cooperative Agreement PHY-2019786 (The NSF AI Institute for Artificial Intelligence and Fundamental Interactions, http://iaifi.org/). This project developed out of the hackathon at the 2024 Summer School hosted by the Institute for Artificial Intelligence and Fundamental Interactions (IAIFI).

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

# A  REINFORCEMENT LEARNING DETAILS

We adapted DQN from Stable-Baselines3 (Raffin et al., 2021) with a Double DQN action selection rule (Van Hasselt et al., 2016). We train the RL agent for $3 \times 10^5$ total environment steps. We limited the episodes to be of length $10^4$ steps (corresponding to 100 hours in simulation). An MLP of fixed size (2 hidden layers with 64 dimensions each and ReLU activation) was used to parameterize the $Q$-function.

Regarding the environment and training, we list several key implementation choices: At the beginning of an episode, the state is zero-padded to always be of length $K = 5$. We experimented with adding a penalty for allowing the number of cells to increase beyond the initial amount upon termination, but found this was not necessary for successful training. Borrowing terminology from the literature on Atari environments (Mnih et al., 2015), we stack $K$ frames (previous cost values) to form the RL agent's state vector. Although initially we let $K$ be a $\mu$-dependent hyperparameter, we found a constant choice of $K = 5$ to work well across the values of $\mu$ studied.

We find exponentially decaying the exploration parameter $\varepsilon$ to work better than the typical linear annealing (with constant, positive final $\varepsilon$) scheduling. We conjecture that this is because non-greedy actions can be quite detrimental (causing the environment to terminate), and exponentially decaying $\varepsilon$ ensures some exploration continues to occur but with increasingly fewer random actions. To ensure the agent is not overly myopic (especially for such long episodes) we found a large discount factor of $\gamma = 0.999$ (corresponding to an effective horizon of $H = (1 - \gamma)^{-1} = 10^3$) to be helpful. When training the agent, we wait until the completion of one rollout episode, and take as many gradient steps as environment steps have occurred.

## A.1  HYPERPARAMETERS

We find that sweeping over a range of hyperparameters (as shown in Table 2) did not have a significant effect on performance, though for reproducibility we list the final hyperparameters used (for $\mu = 0.7$) below for DQN and FQF experiments, respectively. (N/A indicates the value was not swept over, and fixed to the finetuned value.) The discount factor $\gamma = 0.999$, gradient steps per update (1), frames stacked (5) and buffer size (100,000) are fixed throughout.

Table 2: Hyperparameters for Double DQN

| Hyperparameter | Sweep Values | Finetuned Value |
|---|---|---|
| batch size | $16, 32, 64$ | $32$ |
| exploration rate | $0.01 - 0.2$ | $0.05$ |
| learning rate | $10^{-5} - 10^{-3}$ | $3.60 \times 10^{-4}$ |
| target update interval | $1,000, 5,000, 10,000, 30,000$ | $1,000$ |

Table 3: Hyperparameters for (NoisyNet) FQF

| Hyperparameter | Sweep Values | Finetuned Value |
|---|---|---|
| batch size | $8, 64$ | $8$ |
| exploration rate | $0.02 - 0.2$ | $0.2$ |
| learning rate | $5 \times 10^{-4} - 5 \times 10^{-3}$ | $2.7 \times 10^{-3}$ |
| target Polyak averaging | $0.001, 0.005, 0.01$ | $0.005$ |
| $n$-step TD | $1, 2, 4, 32$ | $1$ |
| entropy coeff. | $0, 0.0005, 0.001, 0.003$ | $0.003$ |

# B   FURTHER EXPERIMENTAL RESULTS

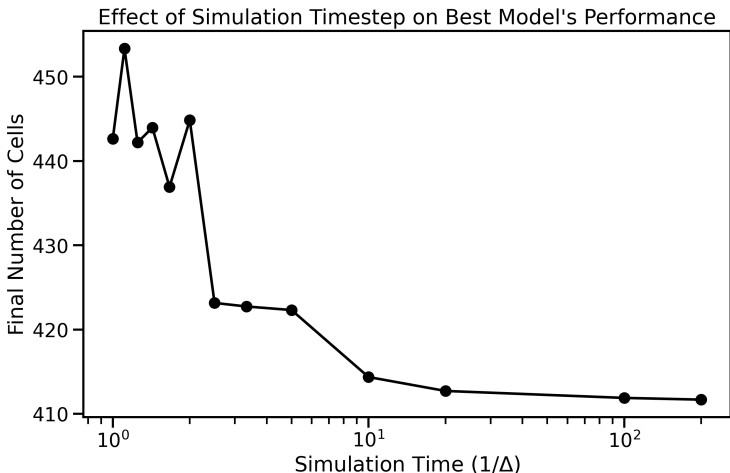

Figure 6: We find that the simulation time can have a significant effect on RL performance. For each choice of $\Delta$, we run a random sweep of size $30$ over various hyperparameters, selecting the highest-performing run for each $\Delta$. Since smaller values of $\Delta$ require longer compute-times for simulations, there is a tradeoff between the amount of time (also, the maximum pulsing frequency, relevant in clinical settings) and the best performance. We have chosen to use $\Delta = 0.01$ throughout.

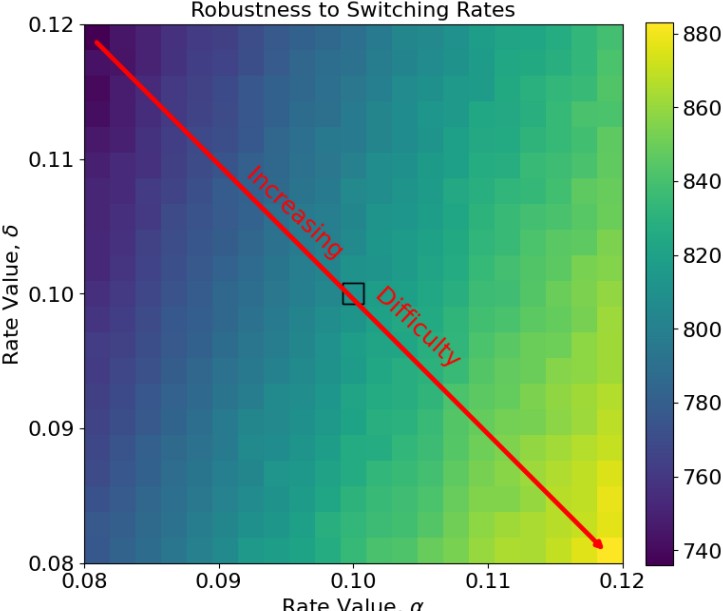

Figure 7: Robustness experiment on changing switching rates $\alpha$, $\delta$. The black box in the center highlights the rates at which the agent was trained. On each axis, the rate is varied by $\pm 20\%$, representing a large range of biologically plausible parameters. As $\alpha$ increases (or $\delta$ decreases), the cells are more often in the resistant state (cf. Fig. 1), making the control problem more difficult for the agent.

