# OpenReview forum: "Reinforcement Learning for Control of Non-Markovian Cellular Population Dynamics"
_ICLR.cc/2025/Conference — ICLR 2025 Spotlight_

### Official Review · Reviewer_JJJP · 2024-10-29

**Soundness:** 4
**Presentation:** 4
**Contribution:** 4
**Rating:** 8
**Confidence:** 3

**Summary:**

This paper studies how non-markovian cellular population dynamics can be chemically controlled through the lens of RL.
While prior work focused on markovian assumptions or unrealistic assumptions, this paper tries to apply deep RL (DQN-based methods) to assess how effective this would be to design drug delivery patterns that optimally control the pathogen cellular dynamics.
The results of the RL-based method are encouraging: the authors show that in the markovian case, the algorithm is capable of finding the analytical solution, and in the non-markovian case a good handling of the cellular growth rate can be achieved.

**Strengths:**

I've found the paper compelling and engaging. It is well written, goes through the necessary background effectively, has enough math to thoroughly back its claims and build the necessary background.

**Weaknesses:**

- The paper mentions that a state encoding the memory is required, transforming the non-markovian dynamic into a Markovian dynamic. The usage of recurrent networks (or perhaps other models like transformers?) could be used to solve that problem without encoding for these states. I find this idea compelling. I wonder if using an RNN is enough material for a follow-up paper, even though it is crucially missing from this one. Hence, not having it within this publication could actually be detrimental for the publicity around this publication altogether.
- Only one family of RL algorithms is being used. Although the authors show that it is robust enough to discover the solution of the problem, it'd be nice to see how DQN compares to other techniques.

**Questions:**

- Why choose DQN and not another algorithm? SAC and others can also be trained in discrete action spaces. It would be nice to have a notion of how DQN compares to other algorithms (in supp. materials).

My main questions have to do with the feasibility of the proposed technique:
- How feasible is it to measure the cell growth periodically in real life settings? I guess one cannot grow cells very frequently and the amount of input data could limit the efficiency of the method proposed. Also, how feasible is it to provide drugs at the proposed rates, and would tighter constraints actually impact the policy much (could it be that a tight constraint on the frequency of the drug administration actually drives the policy to another local minimum?)
- I did not really understand if the plan here would be to train an algorithm offline on simulated data and test it in the real world (sim2real) or if the purpose of this method is to show on simulated data that RL can be used to solve the problem but this will require training the algorithm on real data. Or is the goal to get a standardized drug administration schedule that would fit all cases?
- If the goal is sim2real, some evidence or a path forward to the applicability of this technique would be a nice add-on.
- If the goal is online training, one could maybe consider offline reinforcement learning to help solve this problem?
- How can one assess how good and representative the growth model is? Is there any evidence that sim2real would work?
- Alongside the previous points: is 300K steps for training realistic in real life?

---

> ### Author Response · Authors · 2024-11-21
> **Response to Reviewer JJJP**
>
> We are pleased to read your positive review of our submission. We found your comments and criticisms helpful in further strengthening the manuscript. Below, we provide our responses to your specific questions. Please let us know if this addresses your concerns or you would like to discuss more.
>
> **Weaknesses:**
>
> - Great point: due to the highly non-Markovian nature of the model, we also considered the use of recurrent architectures. Interestingly, our initial experiments showed that using RNN/LSTMs actually gave the same or worse performance, but took much longer to train than vanilla DQN.
> - As a result, we decided to focus on using DQN and FQF given their relative simplicity and robust performance, however we do plan to further study the use of RNNs in future work, in addition to the ideas mentioned in our other responses. As time permits, we can continue to run additional experiments in the same setting with RNNs and report back if there are positive results.
> - As noted in other reviewer responses which raised similar concerns, we agree that another family of RL algorithms should be studied. Thus, we examined the performance of PPO and SAC in this setting. The main results of these experiments are expressed in the general comment at the top of the review thread.
>
>
> **Regarding feasibility:**
>
> - In the experimental setting, measuring the population growth rate of cancer or bacterial cells in vitro periodically is straightforward and performed routinely. Thus, we chose this as our experimental observable (unlike subpopulation size used in prior work, which is harder to measure). As a result, data collection for general model parameterization and testing would not be a major challenge. Since this is where our method would initially be tested, we focused on a realistic experimental setting. Similarly, administration of the drug would require coordination (and potential automation) but there are no major technical hurdles that we can foresee. Implementing precise drug administration and obtaining cell growth rates in the clinical setting at scale would be more challenging, but possible with adequate resources.
> - The main technical limitation in real-world application in the clinic would be obtaining growth rate measurements at short intervals from real patients. Your intuition on various $\Delta$ values driving the optimization procedure to different regimes is interesting, and merits further experimentation in future work. Interestingly, the results shown in Fig. 6 (appendix) indicate that values of $\Delta \leq 0.2$ all yield similar performances and thus may not be a significant limitation in practice.
> - Furthermore, as it is not practical to obtain large amounts of patient data before using the model to train from scratch, our vision would be to obtain initial experimental data to calibrate the population dynamics environment, train the agent in simulation, then finetune the model on real patient data before or during clinical implementation (essentially sim2real, as suggested). This approach is significantly less expensive (especially compared to measuring subpopulation fractions) in both the required data collection and compute, and also allows for personalized treatment models for individual patients.
>
> Thanks again for your review and we would be happy to discuss further as needed.

---

> > ### Comment · Reviewer_JJJP · 2024-11-25
> > **Final assessment**
> >
> > I would like to thank the authors for thoroughly addressing my concerns in their response. I understand that incorporating a recurrent model may not be straightforward and agree that exploring this aspect further could be a valuable direction for future work.
> >
> > After careful consideration, I am sticking with my initial assessment and score of 8. My decision is largely due to my own limitations in domain knowledge, which prevents me from confidently assigning a higher rating. Nevertheless, I believe the work demonstrates significant merit and contribution.

---

> > > ### Author Response · Authors · 2024-11-25
> > > **Thank you**
> > >
> > > Thanks for your reply. We are pleased to hear our responses addressed your concerns and that you believe our work is suitable for publication. We appreciate you taking the time to review our work.

---

### Official Review · Reviewer_9huZ · 2024-11-04

**Soundness:** 4
**Presentation:** 4
**Contribution:** 3
**Rating:** 8
**Confidence:** 3

**Summary:**

The paper addresses the challenge of controlling and driving cell populations, such as cancer cells, toward extinction in environments where exhibiting adaptability and memory of past stressors. This is particularly difficult in systems with unknown parameters or those displaying non-Markovian dynamics such memory-based systems. The authors propose a new memory-based model for switching population dynamics, develop an optimal solution for the memoryless case, and apply model-free deep reinforcement learning (RL) to develop effective drug dosing strategies for cell populations with memory. The proposed approach is tested by comparing its performance against known exact solutions in controlled scenarios, demonstrating that deep RL can accurately recover these solutions. Additionally, the approach is evaluated in more complex environments where the memory strength of cell populations varies. The results demonstrate the method’s robustness in managing dynamic, memory-influenced systems.

**Strengths:**

- Very clear exposition, introduction to the problem and model development
- Contribution of a novel model useful for RL-based optimization of drug dosing
- Useful insights from control theory that allow simplification of the problem
- Optimal solution given for the memoryless case which serves as a sanity check for the RL-based method
- Convincing results of the RL-based solution compared to three baselines
- Clinically highly relevant problem overall

**Weaknesses:**

- Limited methodological contribution: author's mainly apply existing RL methods to a newly developed simulation model (I see the model itself and the insights gained about it as main contribution)
- Not clear whether assumed time-scales are realistic

**Questions:**

- Given that mammalian cancer cells are targeted as an application scenario which generally have a cell division rate of about 24h, how realistic are the assumptions in the paper made about the delta parameter and the control frequencies applied by the pulsating policies?
- Can you give more biological motivation for the experiment with the switching memory strengths in Fig. 4 (right)?
- How important is the switch to FQF from DQN in section 5? How well would DQN do on the more challenging problems considered there? How important is the use of Noisy Nets for exploration over just using annealed epsilon-greedy? These points are not examined, which makes an assessment of the justification for the methodological choices difficult.

---

> ### Author Response · Authors · 2024-11-21
> **Response to Reviewer 9huZ**
>
> We are excited to see that you found the problem and model development exposition clear, and that you appreciate the novelty of the model and of our combination of insights from OC and RL. Furthermore, we are glad you recognize the clinical significance of the proposed problem.
>
> **Methodology:**
>
> Indeed, we agree that the primary goal of this paper was not to develop completely new deep RL techniques. Instead, the main goal was to instead apply existing techniques systematically to a novel, highly-non-Markovian environment, with significant potential real-world impact. The application of RL and OC as presented provides new insights to an otherwise intractable model. As you point out, the model itself and insights gained are part of the main contributions. Furthermore, we believe our approach of using control theory to simplify the RL problem is novel in the field, and we show that combining insights from both can be powerful. As such, we maintain that our work is a strong fit at ICLR, despite not presenting a specific new deep RL algorithm. Please see our response below regarding the timescales used in our simulations, and let us know if you would like to discuss further.
>
> **Questions:**
> 1. We appreciate your concern for the biological plausibility of our model in addition to considering the mathematical and computational aspects of our work. You are correct that individual mammalian cells typically divide roughly every 24 hours; but in a given population, divisions occur asynchronously, so population growth appears much more continuous. Furthermore, cell death and drug uptake can occur at any time and in many cases are not directly connected to division timing. Thus, drug pulsing can affect cell physiology on smaller timescales than the cell cycle. The main real-world bottleneck to a small $\Delta$ parameter is being able to precisely administer the drug at the appropriate time. In the future we hope to collaborate with clinical groups to validate our computational approach in real-world settings, but the focus of the current work is to first establish a computational framework.
> 2. Given that adaptive living systems such as cancerous or bacterial cell populations can dynamically adapt their memory capacity over time, Fig. 4 was aimed at testing the RL model’s robustness to changes in memory strength which could occur as a result of several biological mechanisms such as mutations and changes in gene expression. We have updated the manuscript to make the justification of this experiment more clear, along with additional references giving experimental justification.
> 3. To choose the specific method ultimately employed (FQF with NoisyNets), we swept over different algorithmic choices (with and without PER), specifically: DQN, DDQN, C51, FQF, FQF+dueling, FQF+noisy and chose the model with the absolute best evaluation performance. This model was FQF+noisy (without PER). To add clarity to this point, we are re-running several experiments with finetuned parameters to compare as time permits. However, we would like to emphasize that the main point is to achieve the best evaluation reward, measured as the lowest simultaneous cell counts across $\mu \in [0.6,\ 0.7,\ 0.8,\ 0.9,\ 1.0]$.
>
> Thank you for reviewing our work. We would be happy to continue discussing during the remaining rebuttal period.

---

> > ### Comment · Reviewer_9huZ · 2024-11-27
> > **Assessment after rebuttal**
> >
> > I thank the authors for addressing my points in their rebuttal which was very helpful for me to better understand various design choices and motivations. Correspondingly, I raised my score to an accept.

---

> > > ### Author Response · Authors · 2024-11-27
> > > **Thank you**
> > >
> > > Thank you again for reviewing our work. We are glad to hear that we have fully addressed your points and we greatly appreciate your updated score.

---

### Official Review · Reviewer_yE9z · 2024-11-04

**Soundness:** 4
**Presentation:** 4
**Contribution:** 4
**Rating:** 8
**Confidence:** 3

**Summary:**

The paper proposes to model the non-markovian process of adaptive cellular growth dynamics in bacterial and cancerous populations to learn optimal drug dosing using reinforcement learning. It shows successful training of an improved policy for memory-based systems compared to optimal control, which is robustness to observation noise and memory strength perturbations.

**Strengths:**

The paper is well-motivated and regards a relevant real-world application. It is generally well-written, explained, and easy to follow. It provides a formally sound theoretical connection of the problem at hand to optimal control theory, extended to the application of reinforcement learning (including realistically obtainable observations) to tackle current shortcomings. Convincing empirical results show improved adaptability and generalization to varying memory levels of the underlying model.

**Weaknesses:**

While, in general, the problem is well-explained and elaborated, an example or illustration might have been useful. Also, the figure placement and referencing should be improved to further ease readability. The overall objective of reducing the population by minimizing growth to negative growth could be stated more clearly in the former sections (e.g., 2.1). In Fig. 2 (right), a comparison to the memory-less or optimal control approaches used as a baseline later on could have been insightful.

Minor comments:

- p.2 l.89 no middle part in Fig. 2 (referencing the wrong figure?)
- p.6 l.308 RL uses gradient ascent on the reward, not SGD.

**Questions:**

Do the authors have any intuition on how RL would perform in a continuous drug control scenario?

If the constructed model might not fully describe the underlying biological dynamics (cf. p.4 l 214f.), causing OC to be inaccurate, how does RL solve this issue?

How are the objectives of maximizing net death and maintaining the stability of the subpopulations connected?

Given that learning an RL policy in a specific real-world application would not be feasible (similar to OC), how well does the resulting policy generalize regarding the parameters underlying the assumed model (beyond the memory size)? How can overfitting to the chosen training specification be prevented?

---

> ### Author Response · Authors · 2024-11-21
> **Response to Reviewer yE9z**
>
> **Presentation:**
>
> We are glad to hear that you found the paper well-explained, well-motivated, and relevant. We also appreciate your input on how to further improve readability. As suggested, we have re-done the figure referencing (by fixing a ref to Fig 1 and moving Fig 4 further up) to improve the readability – please let us know if you see any other areas where the readability or referencing can be improved. Additionally, we added a gray dashed line indicating the optimal solution from the memoryless case in Fig 2 (right). Thank you for this suggestion, we agree this provides a better intuition of how the RL solutions compare to that of OC.
>
> **Minor Comments:**
>
> Thank you for carefully reading the manuscript to find these points. We have updated the reference to Fig 1, rather than Fig 2 as you pointed out. Regarding SGD vs. gradient ascent, we only meant SGD is performed on the parameters of $Q_\theta$ based on the Bellman residual (Eq 17). But you are correct about RL in general performing ascent on the return objective. If you think it will help to add a clarifying sentence along these lines, we can do so.
>
>
>
> **Specific Questions:**
>
> 1. Our intuition was that continuous action algorithms would have a more difficult time learning bang-bang control for two reasons: (1) The corresponding bang-bang action values are at the extreme limits of the output policies (thus they tend to saturate gradients at the output layer) and (2) Since neural net representations of the policy implicitly assume a smoothness with respect to state and in some algorithms (SAC) give bonuses for higher entropy, a deterministic policy with such extremes w.r.t. observation would be hard to learn. Based on our recent experiments with SAC and PPO, our basic intuition seems to hold up, and it appears that such algorithms do have a harder time learning in these environments.
>
> 2. We appreciate your concern for ensuring that our RL approach is robust to potentially different or unobserved biological dynamics. As summarized in the general response, our new robustness experiment shows the RL agent performs well across a range of the most important switching parameters, even when they were not seen during training. Moreover, the generalization of our bang-bang derivation to allow for unique dose-response functions for each rate ensures our use of the binary action space for the RL agent is well-founded. Taken together, these results support the notion that our discrete action RL approach will be successful even if our model does not precisely capture the true underlying biological dynamics.
>
> 3. While the objective of maximizing net death rate is indeed the one used in defining the reward function, the goal is to (equivalently) minimize or suppress the total population. To accomplish this, the agent must ensure that resistant cells are periodically driven back to the susceptible state (by withholding treatment), such that the overall population can continue to be reduced. If we mentioned at some point ‘maintaining the stability’, this is not entirely accurate, and we apologize for any confusion.
>
> 4. We foresee this being an important problem for future work, which requires more computational resources and potentially real-world data. In the future, we would like to use known techniques such as domain randomization and recurrence (to have a better internal state representation in a contextual setting) to allow a very general RL policy to be useful in deployment. To avoid overfitting, we foresee the training of an agent in simulation on a vast range over all physically plausible environments, to make the final policy robust rather than overfitting on a specific parameter setting.
>
> Thank you for your careful review, and we look forward to discussing further as needs be.

---

> > ### Comment · Reviewer_yE9z · 2024-11-27
> > **Rebuttal Response**
> >
> > Thank you for your extensive rebuttal and for providing a significantly improved revision, which addresses all of my open questions and the initial weaknesses of the paper I pointed out.

---

> ### Author Response · Authors · 2024-11-27
> **Thank you**
>
> We are glad to hear that our rebuttal addressed all of your questions and concerns. We believe that your feedback did indeed help us improve our manuscript. Please let us know if you have any other questions or concerns, and we would be happy to discuss for the remainder of the rebuttal period.

---

### Official Review · Reviewer_mj4C · 2024-11-05

**Soundness:** 3
**Presentation:** 3
**Contribution:** 2
**Rating:** 6
**Confidence:** 4

**Summary:**

The paper considers the problem of controlling cell-population by adjusting drug dosage. In line with medical research, the authors propose a dynamics model with memory (non-Markov) for this setting, based on ODEs, and derive the theoretical optimal controller for drug administration, that is found to be a "bang bang" controller. Using this insight, the authors formulate this problem as an RL problem, then use RL with reduced action-space to solve it. The method is evaluated against a few naive baselines on a simulated environment, showing both superiority over the baselines, robustness to noise and changes of the memory of the system.

**Strengths:**

### Presentation:
The paper is well written, and for me, it was easy to understand, in addition, it cites many relevant papers across all fields (RL, control theory, medicine).

### Methodology:
I think this paper bridges some of the gap between real-life problems to RL; while it is more applicative in terms of RL, it suggests how to do it right and efficient. The analysis in Sec. 2 enables RL to solve the problem (still need to show it). The experiments are extensive and show results for a few settings, and how it affects the model results.

**Weaknesses:**

### Presentation:
In the current form, the paper is written as two papers: One that formulates the problem and another that utilizes RL to solve the problem. For my opinion, it should have a more solid story, which would require some rephrasing. Considering the title, I suggest noting the difficulties of using RL for the raw problem, and then present your analysis for the optimal control which provides a solution which implies a much simpler problem for RL.

### Experiments:
I think that the naive baselines are not enough here, maybe add an RL variant that operates over the continuous action-space, showing the importance of your optimal control analysis?
Also, there are missing details for the experiments -- for how long the RL agent is trained? what

### Contribution:
My concern here, is that the method is motivated by real world, but eventually tested in simulation. Which shakes the ground of this paper, as it all revolves around the suggested dynamics model. I think another step towards bridging RL and real-life should be added. Since it is less likely to perform experiments that involve real-life environment and control, I would suggest showing empirically that the proposed dynamics model could indeed predict cell population -- using real-life data.

**Questions:**

1. How many random seeds/trajectories did you use in each experiment? What is the faded area in Fig. 4?

2. Have you tested how robust is the resulted policy to parameter noise of the model?

---

> ### Author Response · Authors · 2024-11-21
> **Response to Reviewer mj4C**
>
> **Presentation:**
>
> We are glad to hear that you found our submission well-written, and we appreciate your suggestions to further improve readability. We agree that the presentation of our manuscript would be strengthened by adding the structure that you proposed, in particular by better connecting the OC insights with the RL results. To this end, we have added several lines in the introduction outlining the paper’s structure with the proper emphasis you noted (end of Sec. 1). Furthermore, we believe the addition of continuous action experiments (PPO and SAC, discussed in the general response) successfully ties the two sections together, as they highlight the importance of simplifying the control problem with a discrete action space.
>
>
> **Experiments:**
>
> We are glad to see that you found our experiments extensive and that our work bridges some of the gaps between real-life problems and RL. As we mentioned in the general response, to further strengthen our results, we ran similar experiments with PPO and SAC. Our results indicate that these continuous action algorithms struggle to learn an optimal policy, while attempting to reach a bang-bang control. We added this figure to the paper (new Fig. 4) to further motivate the need for the discrete action space. These experiments independently validate the use of the binary action space derived by OC.
>
> Additionally, we previously included all experimental details, including how long the RL agent was trained ($3 \times 10^5$ steps) in the appendix, but we moved this to the main text for clarity. Please let us know if you believe any other experimental details should be added or moved to the main text.
>
> **Contribution:**
>
> We thoroughly agree that testing on real experimental data would strengthen the paper. However, given the high cost of such real-world data, this work instead develops a principled model (cf Sec 2.1 for references with similar models in other work) demonstrating the applicability of RL to this domain. A goal of this work is to show a strong proof of concept, which will motivate experimental works on adaptive therapy. Furthermore, through Figs. 5 and 7 we show the generalization abilities of deep RL, which offers flexibility even when real-world dynamics differ from simulations. This work paves the way for offline-online finetuning methods when real-world data becomes available in the future.
>
>
>
> **Specific Questions:**
> Thanks for catching this. We have now added it to the caption. Each line represents the mean over 10 trajectories, with the standard error indicated by the shaded region.
> As mentioned above, we just finished running additional experiments to test robustness. The results are summarized above, but please let us know if you have further questions.

---

> > ### Comment · Reviewer_mj4C · 2024-11-27
> >
> > I thank the authors for their response, and for the implemented revisions. I think the paper is much better now. Regarding the contribution - I accept your explanation, although a small discussion on the implications of working with a simulator can contribute. For example, would bang bang control be the form of the optimal policy in the real-world?
> >
> > Nevertheless, I update my score to reflect the revisions.

---

> > > ### Author Response · Authors · 2024-11-27
> > > **Response to Reviewer mj4C**
> > >
> > > Thank you for updating your score and providing this valuable feedback.
> > >
> > > We would be happy to add a discussion on the implications of training in a simulated setting: this is an important point as you've noted. Drawing inspiration from the techniques used in [1], we are drafting some additional text for the discussion section, noting the potential approaches to address this sim2real gap.
> > >
> > > From a biological and physical standpoint, bang-bang is very much applicable in the real world. In fact, the discrete nature makes implementation significantly easier, as preparing drug doses of arbitrary concentration can be difficult and costly. Whether it still represents the *optimal* policy in the real-world ultimately depends on the underlying dynamics of the specific biological system. Although we have shown that bang-bang is optimal regardless of memory strength, switching rate parameters, and specific form of monotonic dose-response curve, other factors such as patient well-being, interactions with the host system, and inclusion of penalties for drug exposure/toxicity, could potentially alter the final policy used. We would be happy to discuss this further in the future outlook section.
> > >
> > > [1]: "Solving Rubik’s cube with a robot hand", https://arxiv.org/abs/1910.07113
> > >
> > > We appreciate your active discussion during this rebuttal period. If you have any other questions or concerns, we would be happy to discuss further.

---

> > > > ### Comment · Reviewer_mj4C · 2024-11-28
> > > >
> > > > Thanks for your response. Adding such discussion would be great, especially emphasizing which factors affect the applicability of your proposed method. I encourage you doing so.

---

> > > > > ### Author Response · Authors · 2024-12-01
> > > > > **Response to Reviewer mj4C**
> > > > >
> > > > > Although we are now unable to upload a new PDF, we have added the following text to section 7 ("Future Work"), and we hope that this addresses your concerns:
> > > > >
> > > > > Since preparing drug doses of arbitrary concentration can be challenging, the derived bang-bang control simplifies the practical implementation of our policies. While our results show that the optimality of such policies is robust against memory strength, switching rates, and dose-response curves, their optimality in the clinical setting may be influenced by additional factors such as patient well-being, interactions with the host system, and penalties for drug exposure. Addressing these constraints highlights the need to bridge the "sim-to-real" gap through both novel computational approaches and testing on real-world data.
> > > > >
> > > > > Inspired by its success in robotics applications [1], domain randomization may prove useful in addressing such computational challenges. By training agents in a diverse set of simulated environments with randomized parameters (e.g., switching rates, dose-response effects, noise levels), policies can become more robust for real-world application. With such general agents successfully trained, an offline RL problem [2] with warmstarting [3,4] could be formulated once real-world data is available. We hope that this approach can provide a path toward generalization, thus enabling more reliable application of RL policies in diverse real-world contexts.
> > > > >
> > > > > [1]: Ilge Akkaya, Marcin Andrychowicz, Maciek Chociej, Mateusz Litwin, Bob McGrew, Arthur Petron,
> > > > > Alex Paino, Matthias Plappert, Glenn Powell, Raphael Ribas, et al. Solving rubik’s cube with a
> > > > > robot hand. arXiv preprint arXiv:1910.07113, 2019
> > > > >
> > > > > [2]: Sergey Levine, Aviral Kumar, George Tucker, and Justin Fu. Offline reinforcement learning: Tuto-
> > > > > rial, review, and perspectives on open problems. arXiv preprint arXiv:2005.01643, 2020.
> > > > >
> > > > > [3]: Ikechukwu Uchendu, Ted Xiao, Yao Lu, Banghua Zhu, Mengyuan Yan, Jos´ephine Simon, Matthew
> > > > > Bennice, Chuyuan Fu, Cong Ma, Jiantao Jiao, Sergey Levine, and Karol Hausman. Jumpstart reinforcement learning in ICML 2023
> > > > >
> > > > > [4]: Rishabh Agarwal, Max Schwarzer, Pablo Samuel Castro, Aaron C Courville, and Marc Bellemare.
> > > > > Reincarnating reinforcement learning: Reusing prior computation to accelerate progress. Ad-
> > > > > vances in neural information processing systems, 35:28955–28971, 2022.

---

### Author Response · Authors · 2024-11-14
**General Response**

Thank you to all reviewers for taking the time to carefully review our submission. We noticed several common questions that we will address first in a general comment before replying to individual reviews in the coming days.
1. **Continuous action algorithms.** Most reviewers suggested comparing against continuous action algorithms, which we agree would be an important contribution. Although OC indicates the discrete action case should be sufficient, we agree this result should be checked experimentally in a continuous action setting. Therefore, we are running several experiments with PPO to experimentally test the bang-bang result from OC and check the quality of the solution obtained from this representative continuous action algorithm.

2. **Parameter Robustness.** Several reviewers also mentioned testing the robustness of learned solutions against the underlying model parameters beyond memory strength. We recognize this is an important challenge to demonstrate the efficacy of our method in real-world settings. As a result, we are now running experiments with the trained FQF agent to test its robustness against biologically-relevant parameter perturbations.

We will add new results in this discussion thread as they arrive, and update the manuscript accordingly. We look forward to engaging with the reviewers during this rebuttal period to further strengthen the manuscript.

---

> ### Author Response · Authors · 2024-11-20
> **Update: New Experiments**
>
> The experiments mentioned above recently finished and we are able to report that the overall message agrees with our prior intuition, summarized as: (1) Continuous action algorithms struggle to learn bang-bang control and (2) the deep RL agent is robust to changes in switching rates within an expected range of perturbation. We elaborate further on these conclusions below.
> We invite the reviewers to check the updated manuscript, where Figure 2 and 7 relate to points (1) and (2) respectively.
>
>
> **Continuous action algorithms**
> - We have run both PPO and SAC on a fixed memory strength ($\mu=0.9$) environment, with ~100 days of compute each in hyperparameter search, on par with the finetuning compute timescales from DDQN and FQF.
> - The best model from these sweeps (i.e. the checkpoint with the highest evaluation reward) was then evaluated, and the results are depicted in Fig 2. This figure illustrates an interesting phenomenon where the continuous action algorithms are able to find the “high dose” constant application period at the beginning of the episode, but as the episode continues, both PPO and SAC attempt a short-lived bang-bang policy before defaulting to the average action (0.5 dose) for the remainder.
> - Because of their inability to reach a bang-bang policy, PPO and SAC underperform DDQN by a significant margin. The algorithm and its corresponding total population of cells remaining at the end of the episode are DDQN (560), SAC (785), PPO (790), where lower is better.
> - We believe continuous action algorithms struggle because of the saturating gradient effect at the limits (bang-bang) of action space. Additionally, continuous action algorithms that give bonuses for policy entropy (e.g. SAC) make learning these deterministic policies even more difficult.
>
>
> **Parameter Robustness**
> - To test the robustness of our trained agent against perturbations or shifts in the underlying model parameters (besides $\mu$), we chose to sweep over the switching rate constants $\alpha$ and $\delta$, which determine the rate of change from the susceptible state to the resistant state and vice versa (cf. Fig 1). We chose these rates because they allow for an intuitive control over the difficulty of the control problem. For the parameter regime which we have focused on ($\kappa_S^{max},\kappa_R^{max}>0$, $\kappa_S^{min},\kappa_R^{min}<0$), population reduction is always possible.
> - However, the difficulty of the control problem can be tuned by changing $\alpha$ and $\delta$, thus reducing the space of pulsing frequencies which successfully lead to population reduction. For example, increasing $\alpha$ makes the control problem harder because cells more quickly transition to the resistant state, thus evading drug treatment.
> - We systematically swept over a grid of $\alpha$ and $\delta$ values in the range $\pm 20$ % outside of the (**fixed**) rates used for training. Despite only being trained on fixed rates, the model generalizes well across the range of values tested.
> - The expected increase in difficulty of control is indicated by a diagonal red line in Fig. 7 (increased $\alpha$ and decreased $\delta$) and the agent is indeed seen to struggle more in the most difficult regime. However, the agent’s performance remains nearly fixed along anti-diagonals, showing invariance to the exact rates (only changes to relative rates affect agent performance).
>
> - Beyond these new figures (now Fig’s 2 & 7), we have also updated Fig. 3 (right) with a visualization of the OC solution. We have also made other changes to the manuscript, highlighted with red text. One additional change we’d like to highlight is a generalization of the OC derivation. This generalization guarantees “bang-bang” optimal control when the dose-response functions are unique for each rate, whereas we previously assumed the same functional form throughout.
>
> We are finalizing the individual reviewer responses and they will be posted shortly. Thank you to all the reviewers for suggesting these experiments, as we believe they further enhance the main messages of the paper.

---

### Meta-Review · Area_Chair_DD1J · 2024-12-26

**Metareview:**

This paper presents an approach using RL to control non-Markovian cellular population dynamics, specifically focused on drug dosing strategies for cancer and bacterial cells. The key findings are: (1) development of a memory-based model for switching population dynamics, (2) derivation of optimal control solutions for the memoryless case showing "bang-bang" control is optimal, (3) demonstration that model-free deep RL can recover these optimal solutions and generalize to more complex memory-dependent scenarios, and (4) evidence of robustness to parameter variations and observation noise.

(b) positive

- good theoretical foundation combining optimal control theory with RL
- practical relevance for medical applications
- good empirical validation including comparison to analytical solutions
- good ablation studies and robustness analysis

(c) Weaknesses:

- limited validation beyond simulation environments (real-world feasibility questions around measurement frequencies and implementation)
- Could benefit from more algorithmic comparisons (addressed in rebuttal)
- The connection between theoretical and RL sections could be stronger

The authors managed to address the reviewers' concerns during the rebuttal. I vote for acceptance.

**Additional Comments On Reviewer Discussion:**

The key points raised during the discussion were:

- need for continuous action space comparisons - addressed by adding PPO/SAC experiments showing these methods struggle with bang-bang control
- parameter robustness concerns - resolved through new experiments demonstrating generalization across switching rates
- real-world feasibility questions - authors provided detailed response about measurement practicality and proposed sim2real pipeline
- biological plausibility of timescales - authors justified model assumptions with reference to cellular biology

All major concerns were addressed through new experiments or clarifications. The additional experiments particularly strengthened the paper's empirical validation.

---

### Decision · Program_Chairs · 2025-01-22

Accept (Spotlight)